METHODS

# A method for identifying local adaptation in structured populations

**Isabela do O**[1]*, **Oscar E Gaggiotti**[2],
**Pierre de Villemereuil**[3,4☉], **Jerome Goudet**[1,5☉*]

**1** Department of Ecology and Evolution, University of Lausanne, Lausanne, Vaud, Switzerland, **2** Centre for Biological Diversity, School of Biology, St Andrews University, St. Andrews, Scotland, **3** Institut de Systématique, Évolution, Biodiversité (ISYEB), École Pratique des Hautes Études, PSL University, MNHN, CNRS, SU, UA, Paris, France, **4** Institut Universitaire de France (IUF), Paris, France, **5** Swiss Institute of Bioinformatics, University of Lausanne, Lausanne, Vaud, Switzerland

☉ These authors contributed equally to this work.
* isabela.doo@unil.ch (IdO); jerome.goudet@unil.ch (JG)

**Data availability statement:** All necessary scripts and data to reproduce this study can be found at https://github.com/isadoo/MethodsLocalAdaptation_LogAV.

## Abstract

Species occupy diverse, heterogeneous environments, which expose populations to spatially varied selective pressures. Populations in different environments can diverge due to local adaptation. However, neutral evolution can also drive population divergence. Thus, testing for local adaptation requires a neutral baseline for population differentiation. The classical $Q_{ST}$-$F_{ST}$ comparison was developed for this purpose. Yet, $Q_{ST}$-$F_{ST}$ frequently fails to account for the complexities of population structure because the theory underlying this comparison assumes that all subpopulations are equally related, resulting in inflated false positive rates in metapopulations that deviate from the island model. To address this limitation we use estimates of between- and within-population relatedness to model population structure. Using those relatedness matrices, we infer the between- and within-population ancestral additive genetic variances under a mixed-effects model. Under neutrality, these inferred variances are expected to be equal. We propose here a test to detect selection based on the comparison of these two estimates of the ancestral variance and we compare its performance with earlier solutions. We find our method is well calibrated across various population structures and has high power to detect adaptive divergence.

## Author Summary

Populations of the same species often face different environmental pressures, driving them to adapt locally. However, even in the absence of adaptation, subpopulations can diverge due to random genetic drift and limited migration. Distinguishing between adaptive evolution and random divergence is a central challenge in evolutionary biology. Traditional methods, such as $Q_{ST}−F_{ST}$ comparison, assume equal relatedness among subpopulations — a simplification that rarely holds in complex real-world scenarios,

**Funding:** This study was funded by the Swiss Science National Foundation grant 31003A_179358, 310030_215709 to JG. The funders did not play any role in the study design, data collection and analysis, decision to publish, or preparation of the manuscript.

**Competing interests:** The authors have declared that no competing interests exist.

leading to flawed conclusions. To overcome this limitation, we developed a novel method that incorporates genetic relatedness among subpopulations, leveraging quantitative genetic theory to estimate ancestral additive genetic variances. Our approach provides a powerful tool for testing local adaptation, reliably distinguishing adaptive divergence from drift across a variety of population structures.

## Introduction

Species are generally distributed in heterogeneous environments. Over time, subpopulations exposed to different environments diverge genetically and phenotypically due to genetic drift, natural selection, and limited gene flow. Environmentally divergent subpopulations are likely to be subjected to different selective pressures, promoting their adaptive divergence. At the same time, distant subpopulations are more isolated, leading to their neutral differentiation [1]. Thus, before concluding that the observed difference is adaptive, one must first reject the hypothesis of neutral divergence by establishing a theoretically justified neutral expectation [2].

$F_{ST}$, the fixation index, is a classic measure used to quantify genetic differentiation among populations. For neutral loci, the degree of differentiation between subpopulations measured by $F_{ST}$ depends on the demographic parameters of the population, such as the effective population size and migration rates. When selection is at play, $F_{ST}$ for the loci under selection will also depend on the strength and mode of selection. When selection acting on a locus is similar across environments (e.g. spatially-homogeneous global adaptation; cf. [3]) $F_{ST}$ will be lower at this locus compared to neutral loci. When selection varies across subpopulations (e.g. local adaptation), then the selected locus will have a larger $F_{ST}$ in comparison to neutral loci.

Many traits of adaptive significance, such as morphological [4], physiological [5] and fitness-related [6] phenotypes , are quantitative traits and result from the interaction of multiple genes [2,7,8]. In this context of polygenic adaptation, $F_{ST}$ measured for neutral loci serves as the null expectation. For these traits, the standard procedure is to compare $F_{ST}$ with its quantitative analogue, $Q_{ST}$, which describes the proportion of additive genetic variance between subpopulations relative to the total additive genetic variance (Eq 1) [9]:

$$Q_{ST} = \frac{V_B}{V_B + 2V_W},\qquad(1)$$

$V_B$ represents the between-population additive genetic variance, and $V_W$ is the within-population additive genetic variance. Since $Q_{ST}$ is based on additive genetic variances, its estimation rests on being able to use controlled breeding designs and environmental effects [10]. Under the assumption that the investigated trait is neutral, we expect $Q_{ST}$ to be equal, on average, to $F_{ST}$. If $Q_{ST} > F_{ST}$, it suggests adaptive divergence, while $Q_{ST} < F_{ST}$ may imply spatially-homogeneous global adaptation (c.f. [3]; but note that [9,11] use 'balancing selection' when referring to this scenario).

However, $Q_{ST}$ estimates are subject to significant uncertainty and potential biases. First, ratio estimates, like $Q_{ST}$, can be biased since the expected value of a ratio differs from the ratio of expectations [12]. Second, the number of subpopulations considered strongly affects the reliability of $Q_{ST}$ estimates, with fewer populations sampled increasing the variability and reducing the power of the statistic [13,14]. Because of the high variability of $Q_{ST} - F_{ST}$ comparisons [14] proposed that instead of relying on a direct comparison between the two statistics, one should find where $Q_{ST} - F_{ST}$ falls in the expected neutral distribution of $Q_{ST} - F_{ST}$ values.

Building on this, [15] developed a simulation-based approach to test whether $Q_{ST}$ is consistent with neutral divergence by comparing the observed $Q_{ST}$ – $F_{ST}$ to a simulated distribution of the neutral expectation. This expected neutral distribution is generated using parametric bootstrapping, drawing from a $\chi^2$ distribution following approximations by [16].

While the improvement of the $Q_{ST}$ – $F_{ST}$ method proposed by [15] reduces the false positive rate (FPR), it still assumes an isotropic population structure. For example, when considering instead a stepping-stone model, the *p*-values do not follow the expected distribution under neutrality, resulting in a non-calibrated test [17,18]. As highlighted in [18], the main issue lies in how between-population variance ($V_B$) is estimated in these models. Typically, $V_B$ is derived from a mixed-effects model where population-level random effects are treated as independent, assuming equal relatedness between all subpopulations. This isotropic assumption does not hold for most natural populations, which often have complex genealogical relationships and migration patterns, leading to the lack of calibration reported above.

To address this, [19] proposed a model based on between- and within-population coancestry to estimate additive genetic variance accounting for non-uniform migration and drift patterns. The coancestry between two populations is the average coancestry between all pairs of individuals one from each population, and the coancestry between pairs of individuals is the probability that randomly sampled alleles from these individuals are identical by descent (IBD). [19] method extends the animal model to the metapopulation level, using coancestry to reflect genealogical relationships between subpopulations. They assume that all studied populations trace back to a common ancestral population. By accounting for population structure, their method provides a more accurate estimation of neutral expectations. Nevertheless, [19] solution still relies on a metapopulation model with specific assumptions, the admixture F-model [20], which has been shown to suffer from issues in estimating coancestries [21]. Besides, it is still unknown whether the test Ovaskainen *et al.* [19] developed can be considered calibrated, given that it does not provide a null-hypothesis framework *per se*.

In this study, we introduce LogAV, a new method for testing the null hypothesis of neutral divergence by comparing the log-ratio of two estimates of the same ancestral additive genetic variance: one derived from a between-population effect and the other from a within-population effect. Under a neutral scenario of phenotypic evolution, these two estimates of the ancestral variance should be equal. This constitutes our null hypothesis in a two-tailed test, which would suggest local adaptation if the estimated ancestral variance is larger when derived from the between-population effects than when derived from the within-population effects. Conversely, it would suggest spatially-homogeneous global adaptation (cf. [3]) when the opposite is true.

We evaluate our method (LogAV) along with both [15]'s method and the Driftsel approach [22], which implements [19] ideas, by applying the methods on neutrally evolving simulated phenotypes. We also assessed LogAV's results for traits evolving under divergent selection in different strengths of selection and population structures, considering a range of population structures, including highly non-isotropic configurations. We found LogAV well calibrated for all tested population structures and able to detect selection when present. Finally we show an application of LogAV to a real already published data set by [23] which had previously been analyzed using driftsel

## Description of the method

LogAV is designed to test whether neutral processes alone can account for the observed divergence between related subpopulations of a metapopulation. We assume that all studied subpopulations originated from a single, panmictic, ancestral population, and through a process

of divergence and restricted gene flow, their genetic diversity becomes spatially structured. As shown by [19], it is possible to write an inferential model to estimate the expected ancestral additive variance ($V_{\mathcal{A}}$ - A definition of the relevant symbols used throuhout this manuscript can be found on Table 1) under neutrality. We define the ancestral additive genetic variance $V_{\mathcal{A}}$ as the additive genetic variance in the quantitative trait that existed in the ancestral population. This model provides a baseline for understanding the contributions of neutral processes and selection to the current genetic makeup of each subpopulation. Our approach examines the historical processes that occurred as they diverged from their common ancestor, as well as the ongoing evolutionary process, including current migration patterns and inbreeding.

In quantitative genetics, an animal model is generally used to estimate additive genetic variance. This model partitions phenotypic variance into different genetic and environmental components. To quantify the covariance of genetic effects between individuals, this model depends on the use of an individual-level relatedness matrix. Extending this principle to structured populations requires quantifying relationships between subpopulations as well as between individuals within those subpopulations [19]. Demographic history and spatial metapopulation configuration influence the distribution of genetic variation, which affects additive genetic variance within and between subpopulations. Thus, any test of local adaptation requires that the neutral differentiation baseline used accounts for these effects.

**Table 1. Table describing relevant symbols, their dimensions, and definitions used throughout the method.**

| Symbol | Dimensions | Definition |
|---|---|---|
| $V_{\mathcal{A}}$ | Scalar | Ancestral additive genetic variance. |
| $V_W$ | Scalar | Additive genetic variance within subpopulations. |
| $V_B$ | Scalar | Additive genetic variance between subpopulations. |
| $\hat{V}_{\mathcal{A},W}$ | Scalar | Ancestral additive genetic variance estimated from within subpopulation effects. |
| $\hat{V}_{\mathcal{A},B}$ | Scalar | Ancestral additive genetic variance estimated from between subpopulation effects. |
| $\omega$ | Scalar | Width of the fitness function peak inversely related to the strength of stabilizing selection |
| $\Theta$ | $n_T \times n_T$ | Individual-level metapopulation-wide coancestry matrix. |
| $\Theta_x$ | $n_x \times n_x$ | Block of coancestry values related to population $x$. |
| $\Theta^P$ | $r \times r$ | Population-level metapopulation-wide coancestry matrix. |
| $\hat{\Theta}^P$ | $r \times r$ | Estimated $\Theta^P$. |
| $\Theta_x^P$ | Scalar | Element $x$ of the diagonal of matrix $\Theta^P$ corresponding to coancestry within-population $x$. |
| $\mathbf{M}$ | $n_T \times n_T$ | Block-matrix of individual-level relatedness within subpopulations. |
| $\hat{\mathbf{M}}$ | $n_T \times n_T$ | Estimated $\mathbf{M}$. |
| $\mathbf{M}_x$ | $n_x \times n_x$ | A block of matrix $\mathbf{M}$ describing relatedness of individuals from subpopulation $x$. |
| $A_{i,j}$ | Scalar | Allele sharing between individuals $i$ and $j$. |
| $\bar{A}_s^P$ | $r \times r$ | Matrix of population-level mean allele-sharing of parental generation. |
| $z$ | Scalar | Phenotypic value. |
| $\mu$ | Scalar | Grand meta-population average. |
| $a^P$ | $r \times 1$ | Average breeding values per subpopulation. |
| $a^i$ | $n_T \times 1$ | Individual breeding value. |
| $\text{Log}_{\mathcal{AV}}$ | Scalar | Log-ratio of ancestral variance from between and within-population effect. |
| $n_T$ | Scalar | Total number of individuals. |
| $r$ | Scalar | Number of subpopulations. |
| $i,j$ | Indices | Indices for individuals. |
| $x,y$ | Indices | Indices for subpopulations. |

LogAV estimates $V_{\mathcal{A}}$ in two separate ways: using the between-population and the within-population coancestries. We then compare these two estimates of the additive ancestral variance. We used a method of moment estimator of coancestries [21] to measure *(i)* kinship within subpopulations and infer ancestral variance from within-population variance, and *(ii)* mean kinship between subpopulations and infer ancestral variance from between-populations variance. Since the mathematical framework is constructed to reference the same coalescence tree, it accurately represents the segregation of alleles associated with the same variance. Thus, in the absence of selection, both variances should be equal.

To illustrate the concepts above, let us first consider a scenario involving two subpopulations. In this simple case, we can use $F_{ST}$ to characterize the population structure of our metapopulation to establish the expected relationship under neutrality between the current genetic variances and the ancestral genetic variance. Namely, the proportion of the total genetic diversity described by the diversity between subpopulations is $2F_{ST}$, and the remaining proportion, $1 - F_{ST}$, is the one within subpopulations. Thus, at the limit of low mutation rate, and under neutrality, we should expect that the between-population additive genetic variance ($V_B$) and the within-population additive genetic variance ($V_W$) are respectively (Eq 2):

$$V_B = 2V_{\mathcal{A}}F_{ST}, \tag{2}$$

and (Eq 3)

$$V_W = V_{\mathcal{A}}(1 - F_{ST}), \tag{3}$$

as demonstrated by [24], and earlier by [25].

However, as noted by others before [20,26,27] $F_{ST}$ is a global summary parameter that does not capture differences in demographic histories among subpopulations under complex population structures. These differences in demography could be for example variation in population size, different branching times between subpopulations, and variation in migration rates between pairs of subpopulations. To capture the complexity of more biologically relevant population structures, we use a matrix of coancestries $\Theta^p$ following [19,21].

$\Theta^p$ contains the expected coancestries between pairs of subpopulations relative to the ancestral population (Fig 1). The superscript "$p$" refers to it being a population-level parameter. Element $\Theta^p_{i,j}$ describes the probability of IBD between a random pair of alleles, one coming from subpopulations $i$ and the other from subpopulation $j$. Diagonal elements of $\Theta^p$ describe such probability for any pair of alleles from distinct individuals within a subpopulation. $\Theta^p$ accounts for the non-independence between subpopulations, shared evolutionary history, asymmetric migration rates, and differences in population sizes, addressing limitations of using the global summary parameter $F_{ST}$ in unevenly structured metapopulations [21]. $F_{ST}$ and $\Theta^p$ are related as $F_{ST}$ is a linear function of the average of the diagonal elements of $\Theta^p$ (Eq 4):

$$F_{ST} = \frac{\frac{\left(\sum_{i=1}^{r}\Theta^p_{i,i}\right)}{r} - \theta_B}{1 - \theta_B}, \tag{4}$$

where $\theta_B = \frac{2}{r(r-1)}\sum_{i=2}^{r}\sum_{j=1}^{(i-1)}\Theta^p_{i,j}$ is the average of the off-diagonal elements of $\Theta^p$.

While $\Theta^p$ describes average co-ancestries across and within subpopulations, the **M** matrix Fig 2, [19] describes relatedness between pairs of individuals. As it is general practice, we assume no cross-population breeding was performed, and thus that **M** is a block diagonal matrix this assumption can be lifted and cross-population breeding accounted for in **M**, (see [19]), *i.e.* relatedness for pairs of individuals from different subpopulations are

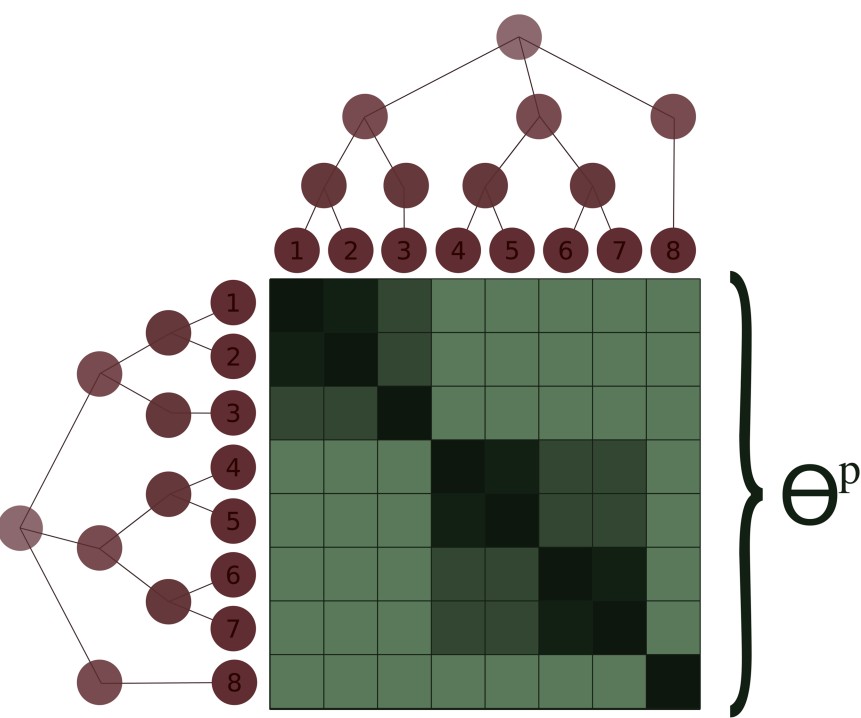

**Fig 1. Cartoon description of $\Theta^p$.** Given $r$ subpopulations, $\Theta^p$ will be a $r \times r$ matrix. In this figure $\Theta^p$ is represented in different shades of green describing the average level of coancestry between random alleles sampled from different pairs of subpopulations, with the diagonal being the average of random pairs of alleles from distinct individuals in the same subpopulation. Darker shades in the cells within the matrix represent higher coancestry between the pair of subpopulations being compared. $\Theta^p$ reflects the past and present demographic metapopulations' histories and connectivity.

already defined with $\Theta^p$, so they are set to zero in **M**. The blocks in **M** describe the relatedness between all pairs of individuals within the subpopulations, relative to the average relatedness within each subpopulation, and discounting relatedness already accounted for in $\Theta^p$. More precisely, **M** contains, within each subpopulation block $x$, the relatedness for each pair of individuals in that subpopulation multiplied by $1 - \Theta^p_{x,x}$.

We develop a test where we estimate, in the same statistical model, the ancestral additive variances considering either (i) the variance in mean phenotype between subpopulations ($V_{\mathcal{A},B}$) or (ii) the additive genetic variance in phenotypes within subpopulations $V_{\mathcal{A},W}$. Under a model of neutral evolution, we expect the two estimates to be equal to the same, unique ancestral additive genetic variance. We thus test the null hypothesis (described in Eq 5 that the two variances are identical:

$$V_{\mathcal{A},B} = V_{\mathcal{A},W}. \tag{5}$$

A scenario where $V_{\mathcal{A},B} > V_{\mathcal{A},W}$ would be compatible with local adaptation, while a scenario where $V_{\mathcal{A},B} < V_{\mathcal{A},W}$ would be compatible with spatially-homogeneous global adaptation.

The use of our method ideally requires phenotypic data of the offspring generation (which we will call F1) reared in a common environment (i.e. "common garden"), neutral genetic markers from the parental ($\mathcal{P}$) generation, and neutral genetic markers or a pedigree for the F1 (we discuss alternatives to this ideal setting in S7 Text). The neutral genetic markers are used to estimate the relative mean coancestry between all pairs of subpopulations and

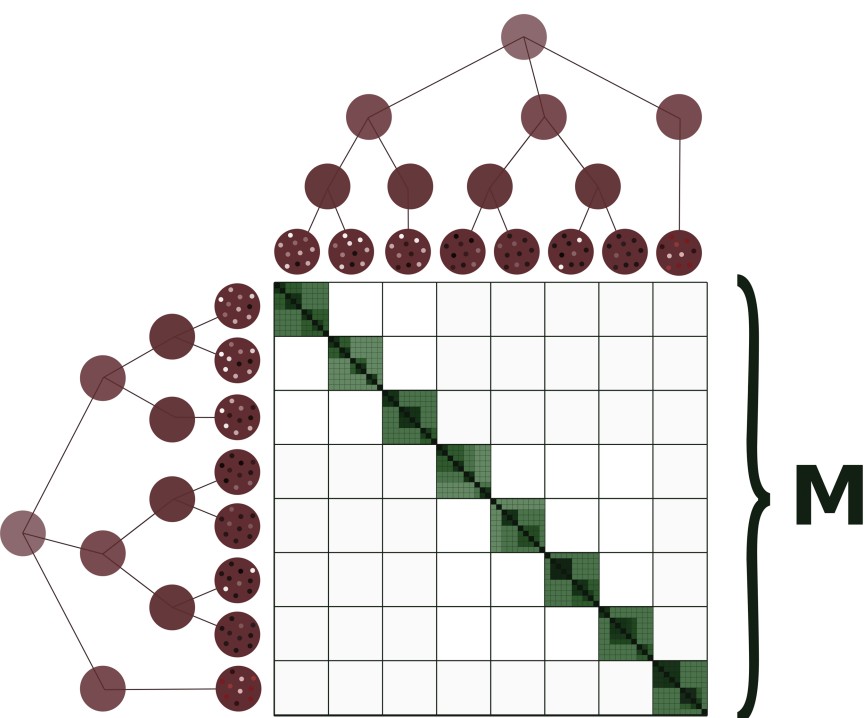

**Fig 2. Cartoon description of $n_T \times n_T$ matrix M.** Similar to Fig 1, darker shades describe higher coancestry, but here, the coancestry is measured for each pair of individuals within subpopulations. Without cross-population breeding in the experimental setup, **M** is a block diagonal matrix.

the relatedness between all pairs of F1 individuals within subpopulations. Fig 3 illustrates at which time point $\Theta^p$ and **M** are estimated and the chosen breeding design.

## Estimating coancestries using the method of moments

We use the allele-sharing method of moments developed by [21,27] to estimate the co-ancestry, using the minimum allele-sharing between subpopulations as a reference point. We use the method of moments to obtain $\Theta^p$ and **M** (**M** can also be obtained using pedigree information if available). The allele-sharing-based estimator can be used for species with any ploidy level and makes no assumptions about the mating system [21]. For a diploid bi-allelic locus $l$, allele-sharing $A_{i,j}^l$ between two individuals $i$ and $j$ takes a value 1 if they are homozygous for the same allele, 0 if they are homozygous for different alleles, and 1/2 if at least one of the two individuals is heterozygous. Over all loci, $A_{i,j} = \frac{1}{L}\sum_{l=1}^{L} A_{i,j}^l$.

To estimate $\Theta^p$, we measure allele-sharing among individuals from the parental generation. The estimate $\hat{\Theta}^p$ is computed as shown in Eq 6:

$$\hat{\Theta}^p = \frac{\bar{A}_s^{\mathcal{P}} - \min(\bar{A}_s^{\mathcal{P}})}{1 - \min(\bar{A}_s^{\mathcal{P}})}, \tag{6}$$

where $\bar{A}_s^{\mathcal{P}}$ is a $r \times r$ matrix made of elements $\bar{A}_{s,(x,y)}^{\mathcal{P}}$, the average allele-sharing $A_{i,j}$ over all distinct pairs of individuals, one from subpopulation $x$ and the other from subpopulation $y$; and $\min(\bar{A}_s^{\mathcal{P}})$ denotes the minimum off-diagonal entry of $\bar{A}_s^{\mathcal{P}}$ [21]. This minimum entry is our

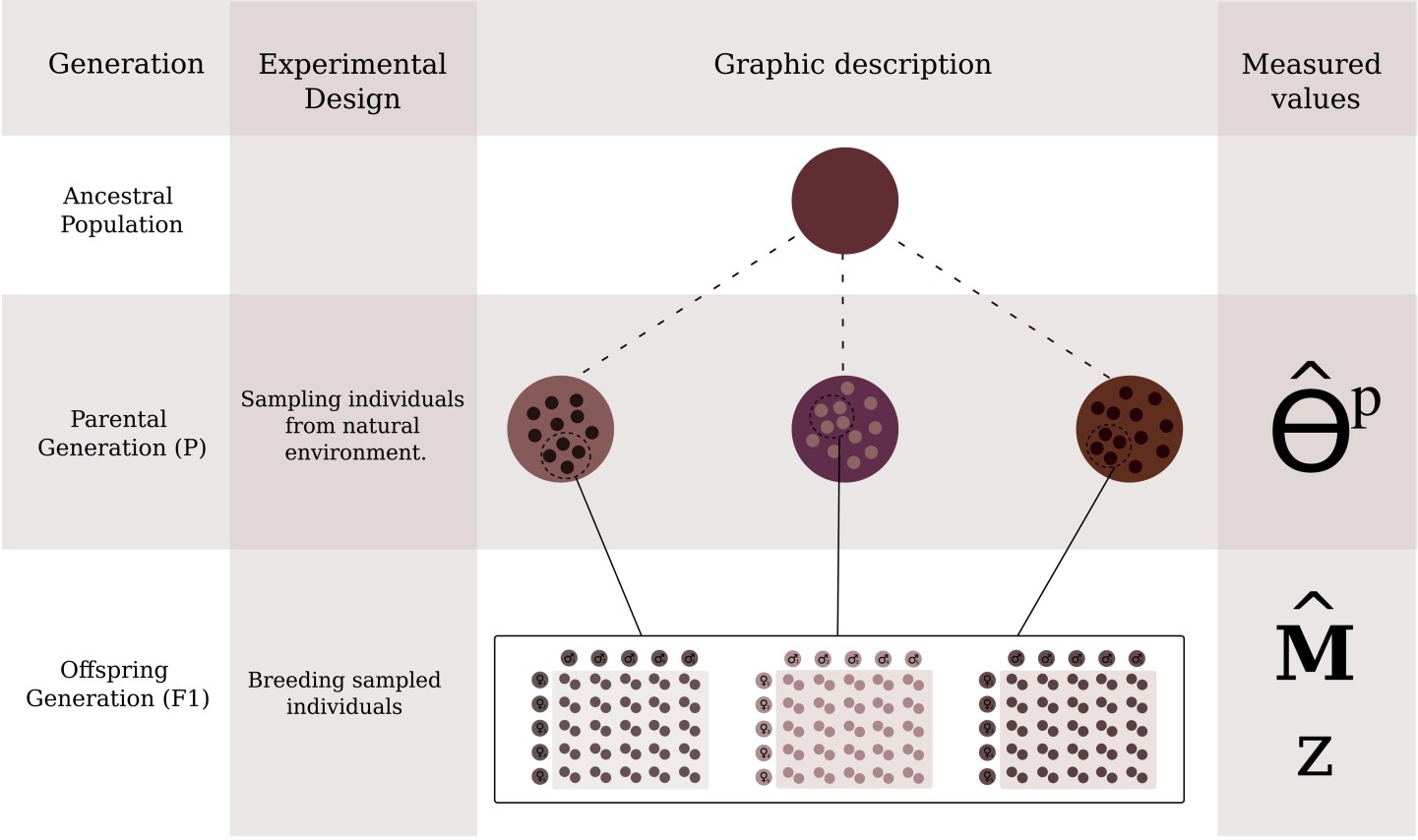

**Fig 3. Description of expected experimental design to estimate $\Theta^p$, M, and z.** The second line of this illustrative table, the "Ancestral population" line, describes the assumed ancestral panmictic state all subpopulations derive from. The third line, "Parental generation" ($\mathcal{P}$) describes the studied subpopulations. From these studied subpopulations, individuals are sampled and exposed to a breeding design. At this step $\Theta^p$ is estimated. On the fourth line, we show the breeding design in the common garden setting used in our simulations: all males, first row of circles, are crossed with all females, second row of circles, following a North Carolina II design (as described by [28]), and two offspring from each cross are phenotyped. We estimate **M** using genotypic (or pedigree information) and phenotypic data from the F1 generation.

best proxy for allele-sharing in the ancestral population, we therefore make the mean coancestry entries of $\hat{\Theta}^p$ relative to the mean coancestry in the ancestral population see [21,29].

Next, we need to estimate **M**, whose elements are the relatedness among the phenotyped individuals ($F_1$) within each subpopulation discounted for their shared ancestries. To obtain relatedness, with pedigrees within each subpopulation, we could simply use the additive relationship matrix, but we can also make use of genotypic information for these $F_1$ individuals if it is available, which we assume here. For each subpopulation, we estimate kinship among all parents and $F_1$ individuals as shown in Eq 7:

$$\hat{k}_{i,j} = \frac{A_{i,j} - \bar{A}_{i,j}}{1 - \bar{A}_{i,j}}, \tag{7}$$

where $\bar{A}_{i,j}$ is the average allele-sharing among pairs of distinct individuals in the subpopulation [30]. We double the kinship to obtain relatedness. To obtain $\hat{\mathbf{M}}_x$, the block of the $\hat{\mathbf{M}}$ matrix corresponding to subpopulation $x$, and to account for the shared ancestry of individuals in subpopulation $x$ relative to the ancestral subpopulation, we multiply the relatedness

estimates of subpopulation $x$ by the scalar $(1 - \hat{\Theta}_x^p)$, where $\hat{\Theta}_x^p$ is the $x^{th}$ element of the diagonal of $\hat{\Theta}^p$. Finally we store all the $\hat{\mathbf{M}}_x$ in a block diagonal matrix (see Fig 2).

### Linear mixed model

We fit a mixed-effects model with the phenotypic trait $z$ as the response variable, accounting for population and individual effects as random-effect components:

$$z \sim \mu + \mathbf{a}^i + \mathbf{a}^p + \epsilon, \tag{8}$$

where $\mu$ is the overall phenotypic mean, $\mathbf{a}^p$ is the population-level additive genetic effect, $\mathbf{a}^i$ is the individual-level additive genetic effect and $\epsilon$ is the residual error. We assume the population-level and the individual-level additive genetic effects follow a multivariate normal distribution with their respective variance-covariance matrices depending on the above-defined co-ancestries and the between- ($V_{\mathcal{A},B}$) and within-population ($V_{\mathcal{A},W}$) ancestral additive genetic variances, whose distributions are described by Eq 9 and Eq 10:

$$\mathbf{a}^p \sim \mathcal{N}(\mathbf{0}, 2\Theta^p V_{\mathcal{A},B}) \tag{9}$$

$$\mathbf{a}^i \sim \mathcal{N}(\mathbf{0}, \mathbf{M} V_{\mathcal{A},W}) \tag{10}$$

Note that assuming different ancestral additive genetic variances for $\mathbf{a}^i$ and $\mathbf{a}^p$ is another notable departure from [19]. We implement this model using a Bayesian framework, yielding notably posterior distributions for the ancestral additive genetic variances ($\hat{V}_{\mathcal{A},B}, \hat{V}_{\mathcal{A},W}$). We used the R package brms [31], based on the Hamiltonian Monte Carlo algorithm from the STAN library [32], with default priors for the fixed and random effects.

### Hypothesis testing
### Verification and comparison

We evaluated our method through simulations conducted in QuantiNemo2 [33], modeling metapopulations evolving under neutral and selective conditions (detailed settings for the simulation are described in S5 Text). Fig 4 illustrates the population structures used for the neutrally evolving metapopulations, along with an example of a coancestry matrix following the expected pattern under neutrality. The simulations were designed to investigate population structures deviating from the standard Island Model, while also varying the number of subpopulations. We chose particularly the 1-dimensional stepping stones to give continuity to the results described in [18] where the authors show the issue with the $Q_{ST}-F_{ST}$ based test on a 20 subpopulation 1D stepping stones. For Stepping Stones we show results for two migration rates, leading to $F_{ST}$ of 0.2 and $\approx 0.37$. The "139" (read: one, three, nine) population structure was inspired by the structure described by [22] and serves as an example of a hierarchical structure. In this structure one population diverges into three subpopulations which each diverges into three new subpopulations, and the data is sampled from the nine subpopulations at the end of the simulation.

In all simulated scenarios, we sampled five males and five females from each parental subpopulation. A common garden breeding design was implemented, wherein the five females from each subpopulation were mated with the five males, producing two offspring per pair. This yielded 25 families of two offspring, thus 50 offspring per subpopulation.

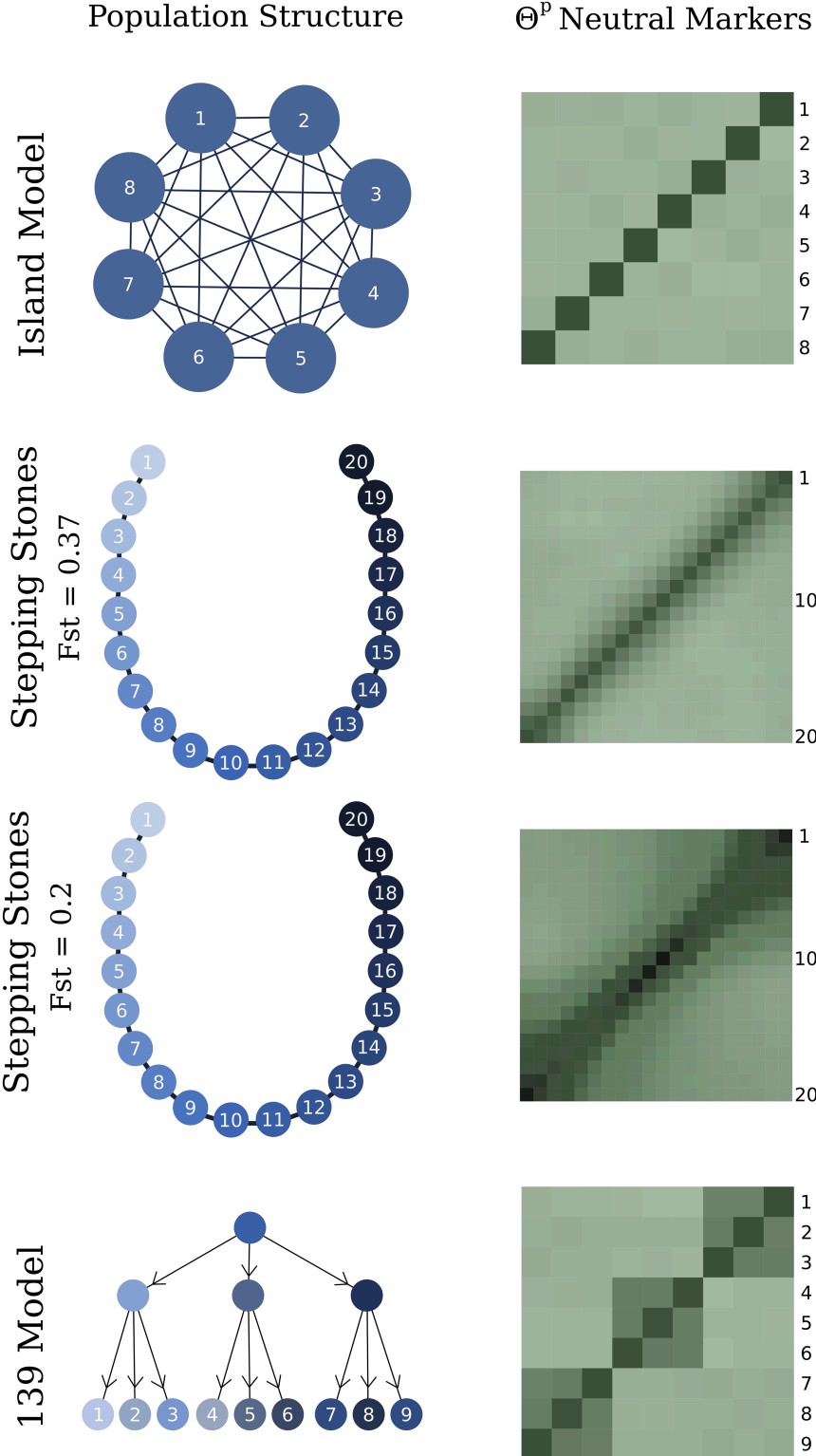

**Fig 4. Description of studied neutrally evolving population structures along with an example of the $\Theta^p$ matrix for neutral markers.** All simulations started from an initial large gene pool that splits instantaneously into 8 subpopulations for the island model and 20 for the stepping-stone model.

Phenotypes were controlled by 100 bi-allelic loci with identical and fully additive effects across all scenarios, while 2000 neutral markers were simulated for genetic analysis. In the selection scenarios, we used stabilizing selection with a different optimum between two halves of the subpopulations, we included three cases of selection for island and for stepping stones model. We were interested in testing which selection intensities were necessary for selection to be identified by LogAV, thus, the selection scenarios differ in the value taken by the parameter $\omega^2$ (see Supporting infromation S5 Text, Eq S3, which describes the stabilizing fitness function). Larger $\omega^2$ describes wider fitness peak, thus a weaker selection, and we studied three cases: $\omega = 10, 22, 50$. These specific values describe a foreign fitness of respectively, 0.6, 0.9, 0.96. In the case of the stepping stones structure, subpopulations sharing the same environment were grouped. The specific parameters for each scenario are outlined in Table 2 and S1 Table.

To compare $Q_{ST} - F_{ST}$ [15] and Driftsel [22] to our method, we focused on the false positive rates (see below), thus we only run these methods over neutrally evolving metapopulations. We analyzed the same set of neutrally evolving metapopulations using the three different methods, $Q_{ST}-F_{ST}$, Driftsel, and LogAV, and compared their results. By analyzing the methods over 500 replicates of each simulated neutral scenario we obtained distributions of $p$-values (in the case of $Q_{ST}-F_{ST}$ and LogAV) and $S$-values (in the case of Driftsel - further details in S4 Text, specifically Eq S1). For our method and Driftsel [22], we used the method of moments [21,27] to estimate coancestry. We made this choice due to results in [21] which show more accurate estimates of coancestry for the method of moments compared to the Admixture F-model (AFM) method proposed by [20].

Under neutrality, $p$-values are expected to follow a uniform distribution, which is also the case for our Bayesian $p$-value due to the asymptotical convergence of the posterior toward the sampling distribution, see [34,35], thus a threshold of 0.05 should correspond to a false positive rate (FPR) of 5%. The FPR is calculated as the proportion of $p$-values falling below 0.05. For Driftsel, under neutrality, $S$-values are expected to follow a normal distribution centered around 0.5 (see S4 Text). Following the criteria outlined by [22], significant $S$-values are defined as being below 0.2 or above 0.8. Although there is, strictly speaking, no theoretical expectation for what the variance of $S$ should be under the null hypothesis, we chose to conform to the values advocated by [22] to measure a "realized" false positive rate when using their method. We performed a binomial test to compare the FPR obtained to the expected one.

**Table 2. Parameters used in the different simulated scenarios.**

| Population Structure | N. subpop. | Subpop. size | Environment | $\omega$ | Optima | Mig. Rate | $F_{ST}$ |
|---|---|---|---|---|---|---|---|
| Island model | 8 | 500 | Neutral | – | – | 0.001 | 0.13 |
| Island model | 8 | 500 | Selective | 10 | -5 or 5 | 0.001 | 0.15 |
| Island model | 8 | 500 | Selective | 22 | -5 or 5 | 0.001 | 0.15 |
| Island model | 8 | 500 | Selective | 50 | -5 or 5 | 0.001 | 0.15 |
| Stepping stones | 20 | 1000 | Neutral | – | – | 0.001 | 0.37 |
| Stepping stones | 20 | 1000 | Neutral | – | – | 0.01 | 0.2 |
| Stepping stones | 20 | 1000 | Selective | 10 | -5 or 5 | 0.0009 | 0.18 |
| Stepping stones | 20 | 1000 | Selective | 22 | -5 or 5 | 0.0009 | 0.18 |
| Stepping stones | 20 | 1000 | Selective | 50 | -5 or 5 | 0.0009 | 0.18 |
| 139 | 9 | 900 | Neutral | – | – | 0 | 0.2 |

## Results

Under neutrality, $p$-values obtained with the log-ratio of ancestral variance method (LogAV) are uniformly distributed for all models of population structure, including those that diverge from the Island Model, and thus yield the expected proportion of significant tests (Fig 5). Conversely, while being correctly calibrated for the island model of population structure, the $Q_{ST}$-$F_{ST}$ method and Driftsel show poor behavior under population structures diverging from the Island Model structure.

As previously shown by [18], $Q_{ST}-F_{ST}$ tests have inflated FPR under Stepping Stone population structures. We find they are significantly different 80 times out of 500 ($p$-value = 2.2 $\times 10^{-10}$) when we expect 25 significant results. Similarly, Driftsel results for Stepping Stone ($F_{ST}$ = 0.35) metapopulations deviate from the distribution expected under neutrality, with inflated empirical FPR (85 significant results out of 500, $p$-value < 2.2 $\times 10^{-16}$). Driftsel also exhibits inflated empirical FPR for the hierarchical structure 139 (44 significant results out of 500, $p$-value = 4.2 $\times 10^{-4}$). For Stepping stones with lower differentiation between demes, *i.e.* a metapopulation with average $F_{ST}$ of 0.2, results are qualitatively similar for $Q_{ST}-F_{ST}$. However, $Q_{ST}-F_{ST}$ tests have even higher FPR. We find significance for 132 replicates out of 500 ($p$-value < 2.2 $\times 10^{-16}$). As of Driftsel, we find lower FPR than the expected 5% in this lower $F_{ST}$ setting, with 15 out of 500 ($p$-value = 0.03). The results for neutrally evolving metapopulations are summarized in Fig 5.

For a further understanding of the differences between the results provided by the different methods we show a graphical comparison between the results obtained using $Q_{ST}-F_{ST}$ and LogAV in Fig S1. Additionally we present the QQ-plots of the theoretical expectation under neutrality for the distribution of statistics (either $p$-value or $S$-statistics) for all three methods in Fig S3 to S5.

We also evaluate the LogAV test ability to detect selection. Under the stepping-stone population structure, the test achieved a 100% success rate across all metapopulations and selection scenarios (Fig 6). By contrast, under the island model, we found that a large peak width of $\omega$ = 50 produced results that closely resembled neutrality. This reduced power is consistent with the similarity in phenotypic distributions between the island model with $\omega$ = 50 and the neutrally evolving population, as shown in Fig S2.

## Applications

We applied LogAV to a previously published data set of nine-spine stickleback from four different populations raised in a common garden [23]. This study analyzed the variation of morphological and behavioral traits using Driftsel [22]. Here, we applied LogAV to the individual traits.

For morphological traits, the original study reports only results from a multivariate analysis, where they observed a signal of diversifying selection. With LogAV, we also find a signal of diversifying selection for four of the traits (Table 3; significant traits are Standard length, body depth, head length, pelvic girdle length).

For the behavioral traits, the univariate results we obtain with LogAV are similar to those reported with Driftsel in the original article, indicating that "aggressiveness" (number of attacks) is likely under selection, while the others ("risk taking 1" and "risk taking 2") could not be shown to be under selection.

## Discussion

In this study, we present LogAV, a novel method based on the log-ratio of ancestral variances to disentangle the effects of selection and neutral evolution on phenotypic differentiation in

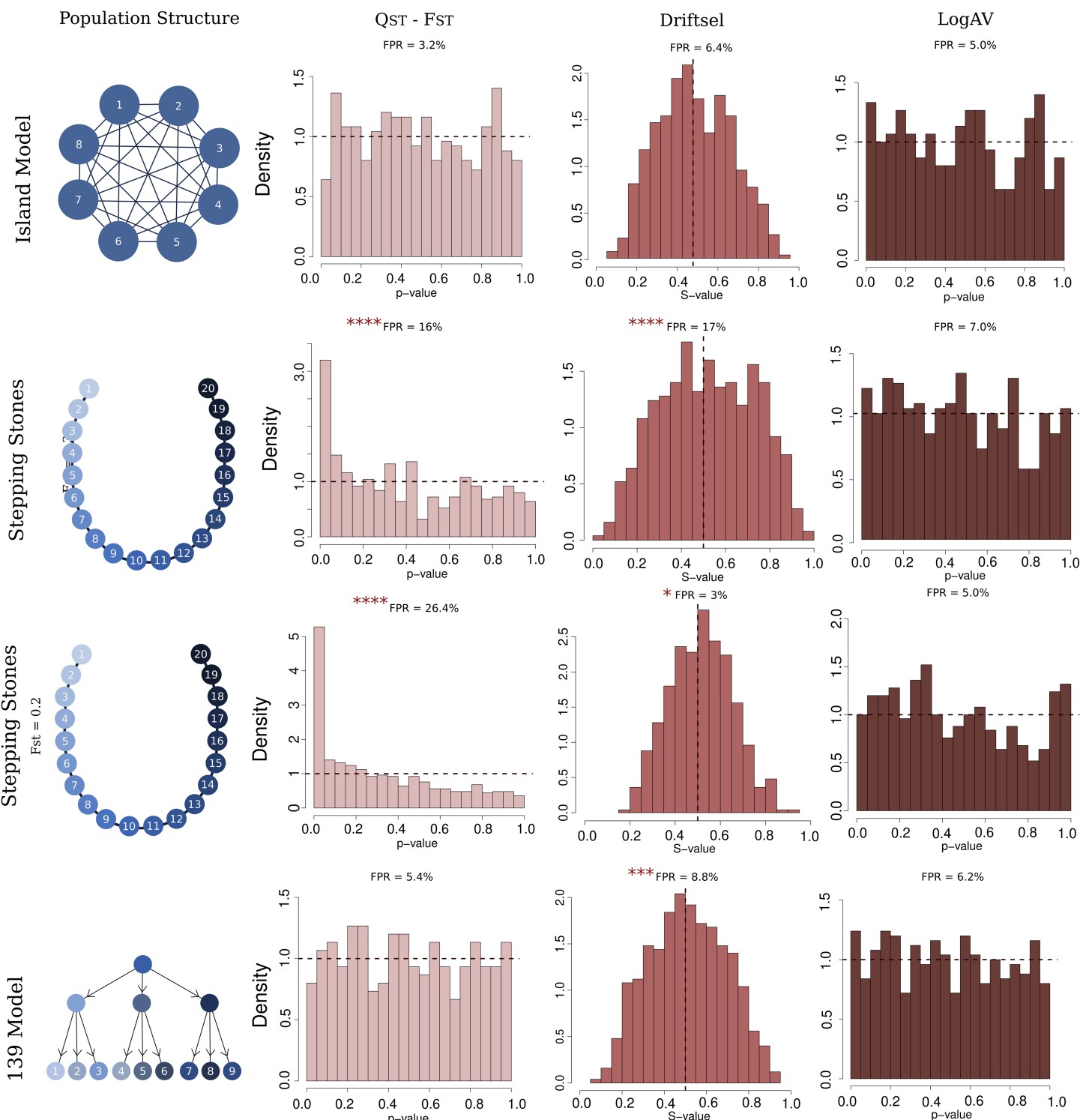

**Fig 5. Summary of the results of analyses conducted on neutrally evolving metapopulations under varying population structures.** The first column specifies the population structures examined. Subsequent columns present the outcomes for each method: $Q_{ST}-F_{ST}$, Driftsel, and LogAV. For $Q_{ST}-F_{ST}$ and LogAV. The results are shown as distributions of $p$-values, while for Driftsel, the results are presented as distributions of $S$-values. False positive rates (FPR) significantly deviating from the 5% expectation are marked with asterisks.

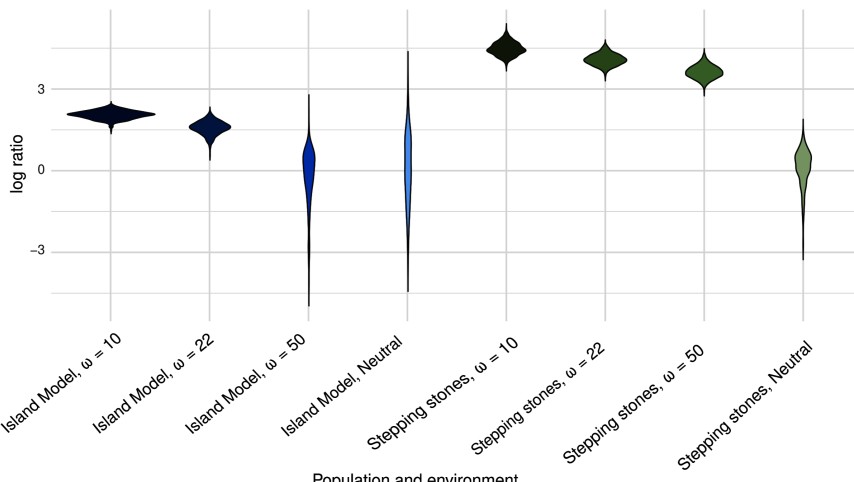

**Fig 6. Distributions of log-ratios from the LogAV test under different selection intensities and population structures.** Each violin plot represents the distribution of $\log(\hat{V}_{A,B}/\hat{V}_{A,W})$ across replicate simulations. We compare two types of metapopulation structure — Island Model (blue tones) and Stepping Stones (green tones) — evolving either neutrally or under selection with different strengths ($\omega = 10, 22, 50$). Higher log-ratios indicate stronger evidence for diversifying selection. The LogAV test reliably detects selection in all Stepping Stones scenarios and in the Island Model scenarios with selection strength corresponding to $\omega = 10, 22$, while it did not detect selection more often than in the neutral simulations for Island Model simulations with weaker selection ($\omega = 50$), where phenotypic divergence is less distinct.

**Table 3. Results of univariate tests of local adaptation using the LogAV method.** For each trait, the table shows the mean log ratio and the associated P-value. Significant evidence for local adaptation (e.g., P-value <0.05) is highlighted.

| Trait | Mean Log Ratio | P-value |
|---|---|---|
| Standard Length (SL) | 4.3955 | **0.0000** |
| Body Depth | 4.8672 | **0.0000** |
| Caudal Peduncle Length (cp.length) | 1.8753 | 0.1210 |
| Head Length | 4.4902 | **0.0000** |
| Aggression (n.attack) | 3.7116 | **0.0488** |
| Pelvic Girdle Length (pg.length) | 3.2546 | **0.0047** |
| Risk Taking 1 (t.fullout)[a] | 0.6225 | 0.6512 |
| Risk Taking 2 (feed)[b] | 2.1483 | 0.1285 |
| Time to Orient (t.orient) | -1.3450 | 0.6328 |

**Note.** Traits were tested individually using the LogAV method. A significant result suggests evidence for local adaptation in the trait.
[a] Time in seconds until the fish fully emerged from a refuge.
[b] Binary response: whether the fish bit food within 300 seconds after a simulated attack.

spatially structured populations. This method builds on the conceptual foundation established by [19], who highlighted the importance of incorporating between-population relatedness to estimate phenotypic differentiation. Through simulations, we demonstrate that LogAV is well-calibrated across all tested population structures. These simulations also showed that existing alternatives such as Driftsel and $Q_{ST}$–$F_{ST}$, are poorly calibrated under neutral scenarios that depart from the island model. LogAV's reliability comes from *(i)* using coancestry matrices computed via the method of moments [21,27], which accurately capture genealogical

ties within and between subpopulations; and *(ii)* using a theoretically justified null hypothesis test framework by comparing two estimates of the ancestral additive genetic variance. This robustness makes it well-suited for real-world use.

Real-world metapopulations often deviate significantly from the assumptions of the island model. For instance, [36] documented a metapopulation of quolls (*Dasyurus hallucatus*) in northern Australia characterized by a hierarchical structure, comprising three main clusters with further subclusters nested within each. Similarly, the bat *Miniopterus schreibersii* demonstrates isolation-by-distance patterns, with additional fine-scale structuring observed within localities [37].

A good example of how structure can potentially impact studies of local adaptation is found in freshwater snails *Galba truncatula*. In western Switzerland, these snails were found in a hierarchical population structure. [38] found even physically close subpopulations, a few meters apart, showed large differentiation, and the differentiation increased as the distances between subpopulations grew. However, the geographic distance alone was not enough to describe the neutral genetic distance, with some far-away subpopulations being undifferentiated, while a small stream was sufficient to create genetic differentiation between physically close subpopulations. Using the same subpopulations, [39] found $Q_{ST}$ to be significantly larger (on average) than $F_{ST}$ between temporary and permanent habitats, but significantly smaller among subpopulations within habitats. It is possible, however, that these results are due to not accounting for the complex population structure in this system.

Studies on local adaptation in a wide range of species have demonstrated that ecological factors strongly influence population structure. For instance, in Daphnia species, differentiation, and genetic diversity strongly depend on habitat size and type [40–42]. Consequently, assuming the Island Model for these species may be an oversimplification, which can lead to high false positive rates in tests of local adaptation. Beyond ecology and life history traits, neutral genetic structure in metapopulations can also be shaped by historical events, such as the recolonization of previously isolated populations following glacial cycles or anthropogenic activities, as is documented for instance in the metapopulations of sticklebacks in western North America [43]. These examples underscore the complexity of natural metapopulation structures, where migration rates and genetic connectivity often vary between pairs of subpopulations. Analyzing phenotypic divergence in such systems requires methods capable of accounting for this structural complexity. Our results show that LogAV is robust to varying population structures and should be a reliable method for testing for local adaptation in the aforementioned species.

In addition to its good performance under neutrality, the preliminary tests of LogAV using traits under selection are promising. So far, we have observed that LogAV demonstrates high sensitivity to selective scenarios. These results were consistent throughout the two different population structures tested under selection. However, further and more thorough tests, especially considering different selection strengths and sample size, are needed to identify its limits.

LogAV should be tested in scenarios that explore the interplay of various parameters, such as migration rates between subpopulations and selection patterns. Tests should include varying both the types of selection patterns and the levels of relatedness, with, for example, closely related populations being exposed to similar selective pressures. It would also be important to assess the method's performance under varying effective population sizes, particularly in scenarios where some populations have much smaller effective sizes than others. We expect LogAV to be robust to such variations, as the inclusion of population coancestries should capture these differences in effective sizes. Additionally, the impact of sampling effort should be examined: for instance, determining the minimum number of phenotyped individuals and

the number of subpopulations. Another point that deserves investigation is how the chosen breeding design will influence the outcome of LogAV [44].

We also suggest further research to test different scenarios of ancestry. The efficacy of the proposed method depends on accurately estimating the ancestral variance from inferred relatedness, which may be impacted by assumptions of panmixia in the ancestral population. Deviations from this assumption, such as in cases of historical population substructure or admixture events, could bias variance estimates. Substructuring of the ancestral population can lead to incomplete lineage sorting and thus to non-uniform coalescent times, which could impact the estimation of the ancestral additive genetic variance. Future research should investigate to what extent the proposed method is affected by the lack of panmixia in the ancestral population.

It is important to mention that other methods have also considered the problem of population structure, and have provided potential solutions considering that little to no information on population of origin and ancestral populations is known. [45] proposed a method for detecting local adaptation using principal component analysis of the relatedness matrix. An advantage of LogAV is the more classic statistical test using an analogue of the *p*-value. [45]'s method depends on the user choosing the threshold of their tests, more specifically the cutoff for which PCs represent between- and within-population variance.

At broader evolutionary scales, other methods have considered similar principles [46,47] as creating neutral expectations from molecular data, and using variance comparisons. [46] proposed a phylogenetic-scale neutrality index by normalizing within and between taxa variance. At a species-level, [47] create a neutral expectation for gene expression evolution by also developing an index but considering sequence divergence and polymorphism in expression data. Still at this broader phylogenetic scale, [48] proposed a framework that bridges statistical genetics and phylogenetic comparative methods. Their work emphasizes how both fields are actually using the same general model, and the importance of considering relatedness when comparing trait associations, whether studying populations or species

Further, we recognize the relevance of conducting additional research to investigate the effect of genetic architecture on the performance of methods for detecting local adaptation. [49] conducted a study that partly addresses this issue by analyzing the impact of genetic architecture on different variations of the $Q_{ST}$–$F_{ST}$. Specifically, they tested the impact of varying numbers of loci impacting the trait on the distribution of $Q_{ST}$. Despite the theoretical expectation of $Q_{ST}$ not depending on the number of loci underlying the phenotype in which the statistic is measured, [49] observed that a higher number of loci leads to higher variance in the $Q_{ST}$ distribution. We have not explored the impact of number of loci, and it is also unclear how dominance and epistatic interactions might impact the results of our method.

Evidence suggests that the classical $Q_{ST} - F_{ST}$ would be uncalibrated when there is dominance. However, the error direction is not resolved. [50] and [51] show that dominance reduces the mean value of $Q_{ST}$, meaning that dominance makes the $Q_{ST} - F_{ST}$ comparison more conservative for divergent selection. While [52] show that under different population structures, $Q_{ST}$ may exceed $F_{ST}$. Since the empirical effect of dominance on genetic variance needs further investigation as some studies show little influence (*e.g.* [53]) of dominance while others show large influence (*e.g.* [54]), it is unclear which effects dominance would have on LogAV. Thus, future research should investigate the extent to which our method could be affected by non-additive effects on the studied phenotype.

A noteworthy difference between our method and $Q_{ST}$–$F_{ST}$ is that it is very general regarding the breeding design used within the common garden experiment. In particular, the $Q_{ST}$–$F_{ST}$ improvements proposed by [15] can be difficult to extend to accommodate breeding designs other than half-sib families. Specifically, their approach is constrained to balanced

datasets, wherein offspring are related as half-sibs through shared fathers. Although [55] suggested an extension to unbalanced datasets, further extensions will be restricted to specific cross or nested breeding designs. Similarly to [19], our method adopts a framework akin to the animal model, which creates flexibility for the breeding design. Further, and extending the flexibility of Driftsel [22], the within-population relatedness matrices can be obtained either from pedigree information or, alternatively, from genetic marker data in the F1 generation, enabling accurate estimation of relatedness [30] even when the exact crosses to generate the F1 are unknown, a situation often encountered for investigations in plants, where seeds are sampled from the field and grown in a common garden environment.

It has been repeatedly suggested in the literature that, by relying on the 'phenotypic gambit' [56], it is possible to alleviate the need for an experimental common garden environment, using direct phenotypic measurements in wild conditions. In such case, a ratio of between-population to the total phenotypic variance, named $P_{ST}$ is computed and compared to the $F_{ST}$ [57]. Our method will then suffer from the same issues as the $Q_{ST}$–$F_{ST}$ does, especially due to the confounding effect of phenotypic plasticity and genotype-by-environment interaction [10,11]. While there exist workarounds for simple occurrences of the first issue [58] and although the phenotypic gambit works to some extent [11], it remains a hardly generalizable simplification. Our method is explicitly based on the assumption that phenotypic traits were evaluated in a common garden settings.

Although the relatedness information used in $\Theta^p$ and $\mathbf{M}$ can be obtained from a wild population settings, the issue of phenotypic plasticity and, especially, genotype-by-environment interaction invalidates our null hypothesis assumption of estimates of the ancestral additive genetic variance, $\hat{V}_{A,B}$ and $\hat{V}_{A,W}$. Our method could possibly be made more robust to such issues by exploring parametrized scenarios of genotype-by-environment interaction, following the spirit of [58], but this would bear the same limits as such approach, and is beyond the scope of this study.

Additionally, we recommend that $\Theta^p$ and $\mathbf{M}$ be calculated respectively from a founder population and from individuals measured in a common garden. While it may be possible to estimate both from the same set of F1 genotypes, doing so may lead to inflated values of $\Theta^p$ due to the added effects of drift across generations in the common garden environment. Potentially, reconstructing parental genotypes from the F1 genotype could be a potential solution [59], though we have not investigated this solution in this study.

In summary, our results validate the log-ratio of ancestral variance as a robust method for distinguishing adaptive divergence from neutral processes in structured populations. Its well-calibrated performance across diverse population structures demonstrates that it provides a reliable alternative to $Q_{ST}$–$F_{ST}$ and Driftsel. We did so by adopting a general framework based on well-established quantitative genetic and statistical principles. From quantitative genetic theory, we derive two distinct approaches to calculate ancestral additive genetic variances, $V_{A,W}$ and $V_{A,B}$. Then we compare the two estimates of ancestral variance using a log-ratio approach that has been adopted by many hypothesis testing frameworks (e.g., likelihood ratio test) and has proven to be very robust in a wide range of applications requiring the comparison of variances.

## Acknowledgments

We thank Otso Ovaskainen and Juha Merila for fruitful discussions and suggestions. We also thank Thibault Latrille, Ehouarn Le Faou, and Michael Whitlock for their comments and suggestions on the manuscript.

## Supporting information

**S1 Text. Explanations for S1 Fig.**
(PDF)

**S2 Text. Estimation of P-values from a Bayesian posterior.**
(PDF)

**S3 Text. Breeding design description.**
(PDF)

**S4 Text. Properties of the $S$-statistic, and why it is normally distributed.**
(PDF)

**S5 Text. Computer simulation details.**
(PDF)

**S6 Text. Quantinemo initialization parameters.**
(PDF)

**S7 Text. Type of data necessary to estimate coancestries and relatedness.**
(PDF)

**S8 Text. Description for S3 Fig.**
(PDF)

**Fig S1. Distribution of p-values of the different methods under neutrality.**
(PDF)

**Fig S2. Distribution of trait values for the different scenarios.**
(PDF)

**Fig S3. Quantile plots for the $Q_{ST} - F_{ST}$ tests.**
(PDF)

**Fig S4. Quantile plots for Driftsel tests.**
(PDF)

**Fig S5. Quantile plots for the LogAV tests.**
(PDF)

**Table S1. Summary of simulation parameters for each scenario.**
(PDF)

## Author contributions

**Conceptualization:** Isabela do O, Oscar E. Gaggiotti, Pierre de Villemereuil, Jerome Goudet.

**Data curation:** Isabela do O, Jerome Goudet.

**Formal analysis:** Isabela do O, Oscar E. Gaggiotti, Pierre de Villemereuil, Jerome Goudet.

**Funding acquisition:** Jerome Goudet.

**Investigation:** Isabela do O, Pierre de Villemereuil, Jerome Goudet.

**Methodology:** Isabela do O, Oscar E. Gaggiotti, Pierre de Villemereuil, Jerome Goudet.

**Project administration:** Isabela do O, Jerome Goudet.

**Resources:** Jerome Goudet.

**Supervision:** Pierre de Villemereuil, Jerome Goudet.

**Writing – original draft:** Isabela do O.

**Writing – review & editing:** Isabela do O, Oscar E. Gaggiotti, Pierre de Villemereuil, Jerome Goudet.

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
