## [Decision Letter · Decision Letter 0]

8 May 2025

PGENETICS-D-25-00294

A method for identifying local adaptation in structured populations

PLOS Genetics

Dear Dr. Goudet,

Thank you for submitting your manuscript to PLOS Genetics. After careful consideration, we feel that it has merit but does not fully meet PLOS Genetics's publication criteria as it currently stands. Therefore, we invite you to submit a revised version of the manuscript that addresses the points raised during the review process.

Please submit your revised manuscript within 60 days Jul 07 2025 11:59PM. If you will need more time than this to complete your revisions, please reply to this message or contact the journal office at plosgenetics@plos.org. Please include the following items when submitting your revised manuscript:

We look forward to receiving your revised manuscript.

Kind regards,

Michael Edge

Guest Editor

PLOS Genetics

Kelly Dyer

Section Editor

PLOS Genetics

Aimée Dudley

Editor-in-Chief

PLOS Genetics

Anne Goriely

Editor-in-Chief

PLOS Genetics

**Additional Editor Comments :**

Three experts have now reviewed the manuscript, and I have read it as well. All readers are agreed that the proposed approach represents a worthwhile addition to the set of methods available for testing for divergent selection among related natural subpopulations, and all are positive about the fit of the paper with PLoS Genetics. At the same time, the reviewers all suggested additions to the manuscript that would make it clearer or more convincing. In particular, reviewers 1 and 2 ask for additional theoretical and methodological detail, and reviewers 1 and 3 suggest additional simulations that would strengthen the manuscript. Reviewers 2 and 3 both ask the authors to consider whether other schemes for estimating pieces of input to the model that might allow for wider application of the method. In a revised version, I would ask that you comment on these possibilities and/or expand on the necessity for the design proposed.

**Journal Requirements:**

2) Your manuscript is missing the following sections: Verification and Comparison, and Applications. Please ensure that your article adheres to the standard Methods article layout and order of Abstract, Author Summary, Introduction, Description of the Method, Verification and Comparison, Applications, Discussion, Acknowledgements, References, and Supplementary Information. For details on what each section should contain, see our Methods article guidelines:

https://journals.plos.org/plosgenetics/s/submission-guidelines#loc-manuscript-organization.

4) We notice that your supplementary Figures, Table, and information are included in the manuscript file. Please remove them and upload them with the file type 'Supporting Information'. Please ensure that each Supporting Information file has a legend listed in the manuscript after the references list.

**Reviewers' comments:**

Reviewer's Responses to Questions

Reviewer #1: Summary

The authors present a simple, robust method to perform QST-FST analyses under a variety of population and breeding structures. They combine and extend previous work on estimating kinship matrices and applying the animal model to study local adaptation. I think that the developed test quantity of the ratio of the between to within population estimates of the ancestral genetic variance is a good idea that will likely apply to a wide range of scenarios. I do not see any major issues with the paper. The structure of the paper is that of advocating a new methodological approach without providing software to apply the method or testing it on real data. I generally think this is fine, but I think that a bit more should be done to justify the approach and identify conditions under which it will fail.

Major comments:

The approach is described in terms of estimating the additive ancestral genetic variance and the provided derivation uses a model of population structure where subpopulations result from one ancestral, panmictic population. However, it is clear that the authors intend for the model to be applicable in scenarios where this is not true. All but one of the simulated models lack any ancestral population. It is therefore strange to see the discussion suggest investigating “to what extent the proposed method is affected by the lack of panmixia in the ancestral population.” The method was tested on three models without any ancestral population at all, and seems to perform quite well, but this is not really discussed or interpreted at all. The authors should do more to at least discuss this generality. In the absence of any software or empirical examples the main focus must be on the theoretical properties.

How should one interpret estimates of the ancestral genetic variance in situations like models 2 and 3? It seems like the ancestral genetic variance may be related to the point at which lineages have been sufficiently shuffled by migration to become exchangeable. Some sort of explanation along these lines would be helpful.

The authors should be more explicit about the variance decomposition leading to equation (8). I was thrown by the seeming possibility for negative values in equation (7) so additional rationale is needed there.

One simple robustness check that could be accomplished with the simulations already performed would be to misassign some individuals to the wrong populations. These may represent recent migrants or incorrect labeling. There is some natural worry with QST methods that populations are correctly defined in the first place. My guess would be that such mislabeling does not lead to false positives since it would inflate within population variance, which the extra between population relatedness would be ignored. This would be good to see explicitly.

Given the focus on the properties of the allele-sharing estimator of coancestry, it would be good to explicitly see whether using other estimators inflates the rate of false positives.

While no software or empirical examples are presented (presumably to be given by future work), it would still be good to know whether the method could be applied to the re-analysis of published data. Driftsel appears to have been applied in at least a handful of published works. Would re-analysis be straightforward?

Minor comments:

144-145: FST is not proportional to the average of the diagonal element of Theta. Doubling this average while maintaining between population relatedness would not double FST?

165: I think stabilizing selection on the phenotypic trait, rather than balancing selection, is meant here.

219: It is not sufficient to say that a Bayesian estimation framework is employed without specifying priors and how the model was fit. I don’t see this in the supplement but it should be there.

269: AFM was not previously defined.

Reviewer #2: In this paper, Isabela et al. developed a new method to estimate the genetic variance between- (Vb) and within-populations (Va) for Qst-Fst comparisons that takes within and between-population relatedness into account. They call this approach LogAV and show that its Type I error is better calibrated than other approaches (e.g. traditional Qst-Fst and DriftSel) for a range of population structures.

The approach makes sense and the authors convincingly show that the errors are well-calibrated. My main comments are related to the lack of sufficient details of the methods and simulations:

I found the details of the inference to be a bit sparse in the main text and supplement. I understand that a Bayesian framework was used to estimate Va and Vb but what were the priors? I assume uniform given their discussion on the equivalence of p-values and posterior probabilities but please make this explicit. What was the range of the prior distribution? Which algorithm was used to sample from the posterior? The code on github suggests MCMC but this and the specific packages used should be explained in more detail in the paper, especially since this is primarily a methods paper.

Please provide more details on the simulation parameters. 100 causal and 2000 neutral markers were used for the simulations. What were their frequencies? Were they drawn from some distribution? Is there a burn-in period before divergence? How exactly were the genetic values simulated to ensure heritability of 0.8 across all scenarios?

Was coancestry estimated from neutral markers only or were the causal variants included? This matters because the frequency distribution and Fst can differ between neutrally evolving markers and those under selection and it's important to evaluate how that would affect estimation of variance components with LogAV.

Fig. S2 is supposed to show 100% success of LogAV in detecting selection. The figure is difficult to interpret beyond showing that the p-values under the neutral scenario are uniformly distributed and under selection to be very low. A better, more informative visualization would be an ROC curve.

Do you strictly need a design where genotypes from the parents are available? My intuition is that it should be possible to get an estimate of theta^p from the offspring generation by methods such as REAP (Thornton et al.) and PC-RELATE (Conomos et al.) to estimate it.

Given that this is primarily a method paper, these details should be made more explicitly available so we can evaluate the performance of the method under different scenarios. It would also broaden the scope of the paper to describe the performance of the method under a wider range of scenarios considering different strengths of selection and genetic architectures. Without this, the paper comes off as more of a preliminary analysis of a promising method as opposed to a fully realized approach.

References:

Thornton, T., Tang, H., Hoffmann, T. J., Ochs-Balcom, H. M., Caan, B. J., & Risch, N. (2012). Estimating kinship in admixed populations. American Journal of Human Genetics, 91(1), 122–138. https://doi.org/10.1016/j.ajhg.2012.05.024

Conomos, M. P., Reiner, A. P., Weir, B. S., & Thornton, T. A. (2016). Model-free Estimation of Recent Genetic Relatedness. American Journal of Human Genetics, 98(1), 127–148. https://doi.org/10.1016/j.ajhg.2015.11.022

Reviewer #3: The manuscript by do O et al explores a new test for selection on quantitative traits by building a linear mixed model and testing for equivalent of the "ancestral population variance" and the within population variance. The test is conceptually similar to Qst-Fst tests, but, as the authors note, expected to be more robust because it can account for more complicated forms of population structure. It also produces something akin to a calibrated p-value distribution, which seems robust to population structure.

I think that the method overall sounds quite reasonable: the authors form two genetic relatedness matrices, one account for the relationship between populations and one accounting for the relationships within each population. They then cleverly use this to build a a statistical test that the within and between population variances are the same. They assess significance using a Bayesian p-value, which is not a concept I was familiar with.

I think that their simulation results, though somewhat sparse, suggest that this method controls false discoveries significantly better than alternative approaches, particularly when population structure deviates from an island model.

While the authors mostly focus on false positive rates, and do a thorough job exploring different scenarios, they only spend a short time (and one supplementary figure) on the power of the test, which they find to be 100% in the scenario they explored. I think that an easy way to make the manuscript have a little more meat would be for the authors to explore a bit more about the power of the method, for example by exploring power across a grid of omega or optimum shift values. It would also be interesting to compare power to the other methods---although given their elevated false positive rate, the authors would need to be somewhat careful about how they frame it.

Another thought that struck me during the reading of the manuscript is that the authors imagine a two step procedure for data collection: first, sampling parents in the field and using them to build the \Theta matrix, and then doing a common garden experiment using the offspring of those samples to measure phenotypes and to measure genotypes to build the M matrix. First of all, it seems like (and perhaps I'm overlooking some aspect of double dipping here) that the same F1 genotypes could be used to construct M and \Theta, which could potentially be useful in some settings?

Secondly, and maybe more interestingly, what if phenotypes and genotypes are measured exclusively from the parental generation? There are many systems in which a common garden is not feasible, but questions of this sort are still interesting. This framework, based entirely on genetic relatedness matrices, seems like it could in principle be applied using purely parental genotypes and phenotypes. Obviously a complication here is that environmental differences between the parental populations will confound the estimation of V_{A,B}. However, it would be interesting to see how strong of an effect would be necessary to really mess up the test. I'm not suggesting that the authors have to do this, as I think he manuscript is already very interesting, but it could be of interest to many practitioners---and anyway, I have a strong suspicion some might just go ahead and do it!

Finally I think that the authors test bears some similarity to a few tests proposed elsewhere in the literature that I think the authors should engage with, e.g. Warnefors and Eyre-Walker (2012 and Latrille et al (2024), which if I recall correctly test a very similar hypothesis about equal variances but motivate it from a McDonald-Kreitman-like perspective. To be clear I absolutely do not think the current manuscript is duplicative of the work in these papers, but they are similar enough that it's probably worth discussing.

I prefer to sign my reviews. My name is Joshua Schraiber.

References:

Warnefors, M., & Eyre-Walker, A. (2012). A selection index for gene expression evolution and its application to the divergence between humans and chimpanzees. PLoS One, 7(4), e34935.

Latrille, T., Bastian, M., Gaboriau, T., & Salamin, N. (2024). Detecting diversifying selection for a trait from within and between-species genotypes and phenotypes. Journal of evolutionary biology, 37(12), 1538-1550.

**Have all data underlying the figures and results presented in the manuscript been provided?**

Reviewer #1: Yes

Reviewer #2: Yes

Reviewer #3: Yes

PLOS authors have the option to publish the peer review history of their article (what does this mean?). If published, this will include your full peer review and any attached files.

Reviewer #1: **Yes: **Evan M Koch

Reviewer #2: No

Reviewer #3: **Yes: **Joshua G Schraiber

**Figure resubmission:**
---

## [Decision Letter · Decision Letter 1]

27 Jul 2025

PGENETICS-D-25-00294R1

A method for identifying local adaptation in structured populations

PLOS Genetics

Dear Dr. Goudet,

Thank you for resubmitting your manuscript to PLOS Genetics. The reviewers from the first round all agreed to re-read the manuscript and have provided their updated assessments. All three are largely satisfied and positive about eventual publication. However, there are some minor issues that require further attention in a minor revision. In a revised version, please address Reviewer 1's requests in regard to ensuring that the available code base is complete, Reviewer 1's comments about figure clarity, and Reviewer 2's request for clarification.

Please submit your revised manuscript within 30 days Aug 26 2025 11:59PM. If you will need more time than this to complete your revisions, please reply to this message or contact the journal office at plosgenetics@plos.org. Please include the following items when submitting your revised manuscript:

We look forward to receiving your revised manuscript.

Kind regards,

Michael Edge

Guest Editor

PLOS Genetics

Kelly Dyer

Section Editor

PLOS Genetics

Aimée Dudley

Editor-in-Chief

PLOS Genetics

Anne Goriely

Editor-in-Chief

PLOS Genetics

**Journal Requirements:**

1) We have noticed that you have uploaded Supporting Information files, but you have not included a list of legends. Please add a full list of legends for your Supporting Information files after the references list.

Note: Supporting Information Legends should not be uploaded as a separate file in the online submission form. 

2) Your current Financial Disclosure states, "JG31003A_179358, 310030_215709Swiss Science National Foundationhttps://www.snf.ch/enThe funders did not play any role in the study design, data collection and analysis, decision to publish, or preparation of the manuscript."

However, your funding information on the submission form indicates receiving no funds. Please modify your funding information.

Please ensure that the funders and grant numbers match between the Financial Disclosure field and the Funding Information tab in your submission form. Note that the funders must be provided in the same order in both places as well. 

**Reviewers' comments:**

Reviewer's Responses to Questions

Reviewer #1: I thank the authors for their responses to my comments and for the changes made to clarify their analysis and improve the manuscript. While the authors did not investigate some of the robustness issues I brought up in my original review, the power simulations and empirical examples are probably more worthwhile additions. The only necessary additions are code and data to reproduce the empirical results. It would also be good to have power simulations of spatially-homogeneous global adaptation, but this is arguably not as interesting as local adaptation.

Code availability:

The GitHub link provides sufficient code and documentation for the generating and analysis of simulated data. I am very happy to see the empirical example where LogAV is applied to the Stickleback data. I would only add that the scripts and ideally data files used to produce Table 3 should also be added to the repository.

Population structure in simulations:

My original confusion with the models of population structure in simulations stemmed from the diagrams in figures 4 and 5 as well as the description in the first paragraph of the “Verification and Comparison” section. No lines are drawn to indicate an ancestral population. In particular, the standard island and stepping stone models of population structure are most often at equilibrium, where the structure persists indefinitely into the past. The authors have pointed out that because of how these simulations are initialized, and the fact that they are not run long enough to reach equilibrium, it is equivalent to having an ancestral population. I don’t expect it makes that much difference to the results, it would be good to explicitly state in the caption to Figure 4. I don’t see any reason not to and it would help clarify things a bit for readers who may naturally assume that such models are the equilibrium versions. I don’t think anything else is necessary here.

Selection simulations:

The addition of Figure 6 is an improvement over what was given in the previous version of the manuscript and the addition of simulations of weaker diversifying selection is welcome as well.

It would have been nice to have matching power simulations under the spatially-homogeneous global adaptation model, but I understand that studies using the method would be more interested in finding local adaptation.

Re-analysis of stickleback data:

I find the empirical example to be a great addition to the paper. This demonstrates that the method will work on real-world data even though the full software implementation is not yet available. I would only ask the authors to make these scripts available.

Reviewer #2: I am overall satisfied with the revisions made by the authors and have no further major revisions to suggest. I would recommend one minor clarification:

Regarding the possibility of estimating theta^p and M from the offspring, the authors give two different reasons for cautioning against this: (i) using all F1 genotypes will lead to inflated theta^p values (in response to my comment), and (ii) inflated theta^p values due to the added drift across generations in the common garden experiment (in the Methods section of the main text). Could they elaborate on their reasoning or provide an appropriate citation? If my intuition is correct, if we have allele frequencies from the sub-populations (e.g. from a reference set), we should be able to estimate the drift in the sub-populations, and therefore, theta^p fairly easily. Could the authors clarify?

Reviewer #3: The revision by do O et al addresses some of my main concerns. I continue to think the manuscript is a little bit terse, but I think the authors added some meat and improved it by adding some more detailed power analysis and by including an analysis of real data. Though the authors did not do any simulations under a model where phenotypes are only measured "in the wild", I think their discussion of the point is acceptable. Regarding the asks in my original review, I feel satisfied by the authors' response.

I prefer to sign my reviews. My name is Joshua Schraiber.

**Have all data underlying the figures and results presented in the manuscript been provided?**

Reviewer #1: **No: **An empirical example using stickleback trait and genetic data has been added but the data and code to generate these results are not present.

Reviewer #2: Yes

Reviewer #3: Yes

PLOS authors have the option to publish the peer review history of their article (what does this mean?). If published, this will include your full peer review and any attached files.

Reviewer #1: No

Reviewer #2: **Yes: **Arslan Zaidi

Reviewer #3: **Yes: **Joshua Schraiber

**Figure resubmission:**
---

## [Editor Report · Decision Letter 2]

4 Sep 2025

Dear Dr Goudet,

We are pleased to inform you that your manuscript entitled "A method for identifying local adaptation in structured populations" has been editorially accepted for publication in PLOS Genetics. Congratulations!

In the meantime, please log into Editorial Manager at https://www.editorialmanager.com/pgenetics/ click the "Update My Information" link at the top of the page, and update your user information to ensure an efficient production and billing process. Note that PLOS requires an ORCID iD for all corresponding authors. Therefore, please ensure that you have an ORCID iD and that it is validated in Editorial Manager. To do this, go to ‘Update my Information’ (in the upper left-hand corner of the main menu), and click on the Fetch/Validate link next to the ORCID field.  This will take you to the ORCID site and allow you to create a new iD or authenticate a pre-existing iD in Editorial Manager.

Yours sincerely,

Michael Edge

Guest Editor

PLOS Genetics

Kelly Dyer

Section Editor

PLOS Genetics

Aimée Dudley

Editor-in-Chief

PLOS Genetics

Anne Goriely

Editor-in-Chief

PLOS Genetics

**Data Deposition**

http://datadryad.org/submit?journalID=pgenetics&manu=PGENETICS-D-25-00294R2

**Press Queries**

---

## [Editor Report · Acceptance letter]

PGENETICS-D-25-00294R2

A method for identifying local adaptation in structured populations

Dear Dr Goudet,

We are pleased to inform you that your manuscript entitled "A method for identifying local adaptation in structured populations" has been formally accepted for publication in PLOS Genetics! Your manuscript is now with our production department and you will be notified of the publication date in due course.

With kind regards,

Zsofia Freund

PLOS Genetics

On behalf of:
